# Generalized Matrix Local Low Rank Representation by Random Projection and Submatrix Propagation

## Abstract

Detecting distinct submatrices of low rank property is a highly desirable matrix representation learning technique for the ease of data interpretation, called the matrix local low rank representation (MLLRR). Based on different mathematical assumptions of the local pattern, the MLLRR problem could be categorized into two sub-problems, namely local constant variation (LCV) and local linear low rank (LLR). Existing solutions on MLLRR only focused on the LCV problem, which misses a substantial amount of true and interesting patterns. In this work, we develop a novel matrix computational framework called RPSP (Random Probing based submatrix Propagation) that provides an effective solution for both of the LCV and LLR problems. RPSP detects local low rank patterns that grow from small submatrices of low rank property, which are determined by a random projection approach. RPSP is supported by theories of random projection. Experiments on synthetic data demonstrate that RPSP outperforms all state-of-the-art methods, with the capacity to robustly and correctly identify the low rank matrices under both LCV and LLR settings. On real-world datasets, RPSP also demonstrates its effectiveness in identifying interpretable local low rank matrices.

## 1 Introduction

Matrix approximation has found wide-range utilities in recommendation systems, computer vision and text mining. Traditional matrix low rank approximation methods, such as truncated singular value decomposition (SVD) and rank minimization, assumes that the observed matrix has a global low-rank, indicating that the low rank components are dense. This becomes challenging in the phase of data interpretation. In real world data, both features and incidences may form sparse subspace structures. As illustrated in **Fig 1**, a matrix can be generated as the sum of a series of local low rank matrices, each consists of a sparse set of features and incidences. One example of such 'locality' property is the purchase history data, where a subset of items were purchased under a common reason by a subset of customers, while neither the items bought together or the users sharing a common purchase reason is known Cheng et al. (2014). Similarly, in biological single cell RNA-sequencing data, a subgroup of genes may be regulated by an unknown signal that is activated only in a subset of cells, which forms a local low rank gene co-regulation module Xia et al. (2017); Wan et al. (2019a); Chang et al. (2020). In addition, shapes, numbers and words in imaging data are also local low rank Lee et al. (2016). In these situations, **M**atrix **L**ocal **L**ow **R**ank **R**epresentation (MLLRR) is more advantageous with its locality assumptions to uncover more interpretable patterns hidden in the data.

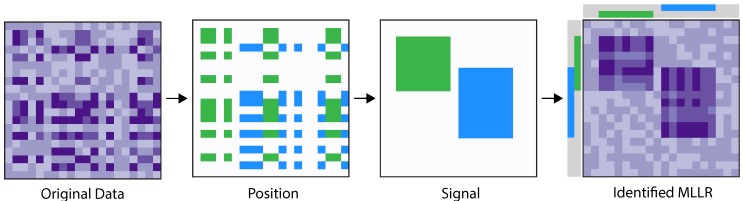

Figure 1: One example of the Matrix Local Low Rank Representation (MLLRR) Problem.

For an input matrix $X \in \mathbb{R}^{M \times N}$, MLLRR aims to identify submatrices $X_{I_k \times J_k}, k = 1...K$, each with a distinct low rank structure, where $I_k$ and $J_k$ represents row and column indices. The total number of possible $I_k \times J_k$ combinations is $2^{N+M}$, making the MLLRR problem NP-hard Stockmeyer (1975). Existing methods for MLLRR fall into three categories: (1) co-clustering approaches that identify submatrices with distinct mean comparing to background Wan et al. (2019b); Banerjee et al. (2007); (2) Sparse matrix decomposition based methods that penalize the number of non-zero entries in the factor/singular matrices Yang et al. (2014); Shen & Huang (2008); Lee et al. (2010); Sill et al. (2011); Witten et al. (2009); Hochreiter et al. (2010); and (3) Anchor-based methods that first pinpoint local regions using certain primitive similarity measure, and further conduct a low rank fitting to each anchored region Lee et al. (2016). Noted, all these methods rely on the assumption that the low rank submatrix has a distinctly spiked mean compared to the background Yang et al. (2014). Identification of submatrices with a more general definition of low rank property remain unsolved, including: (1) the submatrices having a low rank and a similar mean compared to the background, (2) the background noise matrix having heterogeneous and even contaminated distributions, and (3) the submatrices is of a small size or has heterogeneous error distributions.

In this study, we re-visit the tasks of MLLRR from the perspective of random projection and developed a new framework, namely RPSP (**R**andom **P**robing-based **S**ub-matrix **P**ropagation). RPSP first evaluates low rankness of a large set of randomly sampled small submatrices, and gradually grow these low rank submatrices using a propagation strategy. RPSP adopts a random projection approach to approximate the singular values of submatrices that drastically improved the computational efficiency compared to the conventional QR decomposition-based computation.

We systematically benchmarked RPSP with state-of-the-arts (SOTA) methods on comprehensively simulated data and two real-world datasets. RPSP outperformed all SOTA methods on different senarios. RPSP is shown to have the unique capability to handle heteroscedastic error distributions, and distinguish a true local low matrix from background noise with or without spiked mean structure. Application of RPSP on real world datasets demonstrated its capability in detecting context meaningful local low rank matrices.

The key contributions of this work include:
(1) **RPSP is the first general solution for the LCV and unsolved LLR problems**: Compared to existing methods, RPSP is the only method that can robustly solve the MLLRR problem when (i) the patterns are small, (ii) the mean of patterns is not necessarily distinct compared to background, (iii) the background error is non-Gaussian or heterogeneous, and (iv) there are a large of low rank submatrices of different size and ranks.
(2) **A new perspective in analyzing and embedding local low rankness**: We developed new framework for computing and propagating local low rankness of submatrices by estimating singular values of randomly sampled small submatrices and propagating small low rank submatrices to gradually grow into larger ones. This framework directly computes the probability of local low rankness, which also serves an interpretable embedding of matrix data.
(3) **A theoretical framework is developed from mathematical theories of random projection** that support: (i) the identifiability of local low rankness, (ii) bound of sensitivity and specificity, and (iii) impact of errors and pattern sizes with respect to the setting of hyperparameters of RPSP.
(4) **An efficient computation of singular values**: A random projection and parallel computing-based method on GPU was developed to drastically increase the computation efficiency of singular values for a large set of small matrices.

## 2 PRELIMINARIES

### 2.1 NOTATIONS AND MATHEMATICAL BACKGROUNDS

We denote a matrix $X$ of $M$ rows and $N$ columns as $X^{M \times N}$, and its $(i, j)$-th entry as $X_{ij}$. We use $I_k \subset \{1, ..., M\}$ and $J_k \subset \{1, ..., N\}$ to denote row and column indices, and $X_{I_k \times J_k}$ means a submatrix indexed by $I_k \times J_k$. $||X||_1, ||X||_2, ||X||_*$ denote element-wise $\mathcal{L}$-1, $\mathcal{L}$-2, and nuclear norms of a matrix, here nuclear norm is the sum of all singular values of $X$. For $Z \in \mathbb{R}^{M \times N}$, we use $Rank(Z)$ to denote the rank of the matrix. $Rank(Z) = r$ if and only if $Z = UV^T$, where $U \in \mathbb{R}^{M \times r}$ and $V \in \mathbb{R}^{N \times r}$ are two orthogonal rank-$r$ matrices. To say that $Z$ has a low rank property, we mean $r \ll min(M, N)$. Intuitively, a random matrix will have a rank of $min(M, N)$. The low rank property of a matrix added by background noise is commonly characterized by truncated SVD, as defined below.

**Definition 1. Truncated SVD.** Let $Z^{M \times N} = U\Sigma V^T (M \geq N)$ be the singular value decomposition (SVD) of $Z$, where $U^{M \times N}$ and $V^{N \times N}$ are left and right singular vector matrices, $\Sigma^{N \times N}$ is a diagonal matrix of singular values. Define diagonal matrix $\Sigma^{(r)N \times N}$ such that $\Sigma_{ii}^{(r)} = \Sigma_{ii}, i \leq r$; $\Sigma_{ii}^{(r)} = 0, i > r$, i.e., the top $r$ singular values. The truncated SVD of $Z$ of rank $r$ is defined as $tSVD^r(Z) = U\Sigma^{(r)}V^T$. By its definition, the numerical rank of $Z$ is $r$ if and only if $Z = U\Sigma^{(r)}V^T$. When $Z$ is added by noise, we utilize optimal tSVD to characterize the low rank property of $Z$, which has been widely utilized in low rank representation of a matrix with errors Owen et al. (2009); Wold (1978). Specifically, the optimal tSVD of $Z$ is defined as $tSVD^*(Z) = U\Sigma^{(r^*)}V^T$, where $r^*$ is the smallest number such that $E := Z - U\Sigma^{(r^*)}V^T$ that forms an independent and identically distributed noise matrix or any predefined noise structure. We call $r^*$ as the estimated rank of $Z$. Noted, estimating the exact rank $r^*$ requires additional assumption of noise, which is not the primary focus of this work. Nonetheless, a matrix is low rank if and only if $\frac{\sum_{n=1}^{k} \Sigma_{ii}}{||Z||_*}$ ($\forall k \leq r^*$) is substantially larger compared to a random noise matrix.

## 2.2 PROBLEM STATEMENT

**Definition 2. Matrix Local Low Rank Representation (MLLRR).** For a given matrix $X^{M \times N}$, MLLRR identifies $K$ low rank submatrices $X_k := X_{I_k \times J_k}, I_k \subset \{1...M\}, J_k \subset \{1...N\}, k = 1, ..., K$, s.t. $tSVD^*(X_k) = U_k\Sigma_k^{(r_k)}V_k^T, U_k \in \mathbb{R}^{m_k \times r_k}, V_k \in \mathbb{R}^{n_k \times r_k}, r_k \ll min(m_k, n_k)$, i.e., $X_k$ are low rank submatrices added with error. Here $m_k$ and $n_k$ denote the cardinality of $I_k$ and $J_k$.

The MLLRR problem could be categorized into two sub-problems based on the low-rank property: (i) **LCV** (Local Constant Variation) considers submatrices with spiked expected means ($\mathbb{E}$) compared to the background, i.e. $\mathbb{E}(X_{ij}) = u_k, \forall(i,j) \in I_k \times J_k, \mathbb{E}(X_{ij}) = u_0, \forall(i,j) \notin \{I_k \times J_k\}_{k=1}^K$. (ii) **LLR** (Local Low Rank) considers submatrices whose optimal tSVD are of a low rank $r_k$, i.e. $tSVD^*(X_{I_k \times J_k}) = U_k\Sigma_k^{(r_k)}V_k^T$, where $1 \leq r_k \ll min(m_k, n_k)$.

Noted, the LCV problem is a special case of LLR. We separate the two problems because most existing approaches only solve the LCV problem. Let $E_k := X_k - tSVD^*(X_k)$ be the background noise of the $k$th submatrix. Importantly, real world data is often noisy and heteroscedastic, meaning (1) the distribution across different $E_k$ may not be identical; (2) for $(i,j) \notin \{I_k \times J_k\}_{k=1}^K$, $X_{ij}$ may not be identically distributed; and (3) certain entries in $X_k$ may be outliers that could corrupt its low rank structure. Of note, in this study, we do not restrict the form of the background noise distribution.

## 2.3 RELATED WORKS

Currently, there exits three types of methods that can solve the MLLRR problem, namely co-clustering, sparse matrix decomposition, and anchor-based approaches. The main goal of co-clustering is to find a matrix partition such that the intra-co-cluster distance could be minimized, where the distance measure is defined as the Kullback–Leibler divergence in Bregman co-clustering Banerjee et al. (2007), and Euclidean distance in the plaid model Lazzeroni & Owen (2002). For matrix decomposition based methods, they identify local low rank matrices by imposing sparsity constraints to the factor or singular matrices $U, V$ Witten et al. (2009); Yang et al. (2014); Lee et al. (2010); Shen & Huang (2008); Sill et al. (2011). For anchor-based methods, Lee et al. proposed the LLORMA method by using prior knowledge to select anchors of local low rank patterns and their nearby points with a smooth kernel function Lee et al. (2016); Chen et al. proposed the WEMAREC method that builds upon the submatrices identified by co-clustering methods Chen et al. (2015). More details on the existing method formulations were provided in APPENDIX. Among the three types of methods, co-clustering only solves the LCV problem. Although both matrix decomposition and anchor-based methods focus on detecting LLR submatrices, only those submatrices with a distinct spiked mean could be detected. In other words, only the LLR submatrices having the LCV property could be detected. In addition, sparse matrix decomposition tends to detect large submatrix that may explain better the variance of the whole matrix, while sacrificing the locality of the low rankness Yang et al. (2014); Shen & Huang (2008), and the high computational cost of anchor-based methods are not scalable to large matrix Lee et al. (2016); Chen et al. (2015). And none of the existing methods is capable of handling outliers or heteroscedastic errors. In summary, there is lack of an effective and scalable solution for the general MLLRR problem, especially when the mean of the target submatrix is not different from the background and the data contains outliers and/or heteroscedastic errors.

## 3 RPSP AND ITS MATHEMATICAL BASIS

The biggest challenge with local low rank submatrix detection lies in that neither the row or column indices of the submatrix is known. As given in Definition 2, the low rankness property of a submatrix is evaluated through the computation of its singular values, which apparently can't be evaluated until the submatrix has been presented. However, it is computationally impossible to go through all the submatrices of an input matrix. RPSP grows a submatrix of low rank from smaller ones, which utilizes two facts. Firstly, for a low rank matrix $X^{M \times N}$ of rank $r \ll min(M, N)$, the self consistency property suggests that any $M_0 \times N_0$ submatrix $(M_0, N_0 \geq r)$ randomly sample from $X$ is most likely to have a rank of $r$ (Lemma 1 in Owen et al. (2009)). Secondly, for a given matrix, the total number of square submatrices of dimension $M_0$ grows exponentially with $M_0$. The first fact indicates that, any submatrix of low rank is a collage or complete coverage of its own (smaller) submatrices, which are also of low rank. The second fact indicates that the only way for us to grow a local low rank submatrix is to start from the much smaller submatrices. In fact, for $M_0$ as small as 2, it is computationally feasible for us to obtain a full collection of $M_0 \times M_0$ submatrices that could densely cover $X^{M \times N}$. By teasing out all the $M_0 \times M_0$ submatrices of low rank, we could then gradually build them up into larger low rank submatrices. The evaluation of the low rankness for a large number of submatrices now becomes computationally expensive. In RPSP, our biggest contribution is that we have developed a singular value approximation method using random project to efficiently evaluate the low rankness of any given submatrix, making it possible for us to reconstruct a low rank submatrix from its parts.

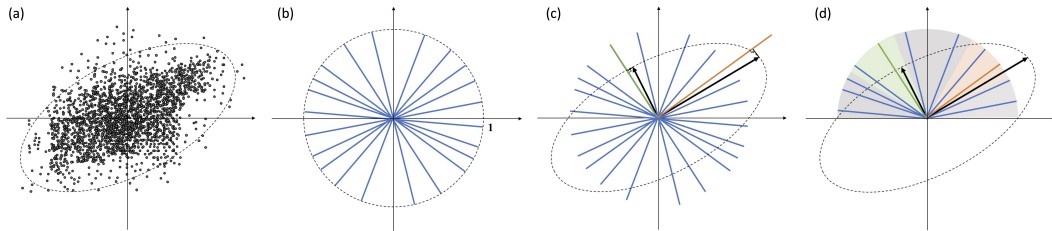

Figure 2: Mathematical considerations of random projection-based singular value computation.

### 3.1 EFFICIENT COMPUTATION OF SINGULAR VALUES FOR A LARGE SET OF SMALL MATRICES

Conventionally, singular values are computed by QR decomposition of $O(n^3)$ complexity. In this study, we developed an alternative approach to drastically increase the efficiency of computing singular values for a large amount of small matrices from the perspective of random projection. Without loss of generality, we illustrate our mathematical bases and the computation of singular values on square matrices. In **Lemma 1**, we first proved that the singular values of $X$ can be estimated via simple operations against a set of randomly sampled unit vectors, including inner products, sum of squares, and max pooling, which can be efficiently computed on GPU. **Lemma 2** derived from the theories in Random Covering of Unit R-Sphere suggests the minimal requirement for densely covering a Unit R-dimensional Sphere. It estimates the number of random unit vectors that are needed to ensure that for any vector in $\mathbb{R}^R$ there almost surely exists at least one random unit vector, whose cosine similarity to the vector is larger than $\cos \theta$.

**Lemma 1.** For a given dimension $R$, denote $X^{R \times R}$ as an input matrix and $P \in \mathbb{R}^{R \times N^R}$ as a matrix of $N^R$ randomly generated unit vectors in $\mathbb{R}^R$. $Y = XP$ denotes a random projection of $X$, then $\lim_{N^R \to \infty} \max_{1 \leq j \leq N^R} \sqrt{\sum_{i=1}^{R} Y_{ij}^2} = \sigma_1$ and $\lim_{N^R \to \infty} \min_{1 \leq j \leq N^R} \sqrt{\sum_{i=1}^{R} Y_{ij}^2} = \sigma_R$, here $\sigma_1$ and $\sigma_R$ are the largest and smallest singular values of $X$. Denote $P_{(1)} = \arg\max_{P_{\cdot,j}} \sqrt{\sum_{i=1}^{R} Y_{ij}^2}$, $\lim_{N^R \to \infty} \max_{P_{\cdot,j} \in Sp(P_{(1)})^\perp} \sqrt{\sum_{i=1}^{R} Y_{ij}^2} = \sigma_2$, where $Sp(P_{(1)})^\perp$ denotes the null (or complemented) space of the linear space spanned by $P_{(1)}$. Similarly, define $P_{(r)} = \arg\max_{P_{\cdot,j} \in Sp(P_{(1)},...,P_{(r-1)})^\perp} \sqrt{\sum_{i=1}^{R} Y_{ij}^2}$, $\lim_{N^R \to \infty} \max_{P_{\cdot,j} \in Sp(P_{(1)},...,P_{(r-1)})^\perp} \sqrt{\sum_{i=1}^{R} Y_{ij}^2} = \sigma_r$, for $r \in 3, ..., R-1$.

**Lemma 2.** The minimum number of caps of half angle $\theta$ required to cover the unit Euclidean R-sphere is called the *Random Covering of the Unit $R - Sphere$*. Then

$$N_c(R, \theta) = exp(R \cdot f_c(\theta)(1 + \epsilon_R(\theta)))$$

, where $\epsilon_R \longrightarrow 0$ as $R \longrightarrow \infty$ and $f_c(\theta) = -log \sin \theta$. Here $N_c(R, \theta)$ is the minimum number of caps with the given dimension R and half angle $\theta$. This means, when $R$ is large enough, if we randomly choose $exp(-Rlogsin\theta + o(R))$ caps, then the area of the uncovered surface of the $R$-sphere will be almost negligible.

The proof of **Lemma 1** is given in APPENDIX and the **Lemma 2** was proven as a Corollary in the section $II$ of Wyner (1967). **Lemma 1 and 2** together suggest that for a given matrix and the level of error to be tolerated, its singular values could be estimated by simple operations against a set of randomly generated unit vectors, which form the mathematical bases of **Algorithm 1: Singular Value Approximation**. Its input includes a matrix $X$ (**Fig 2a**) and a set of randomly generated unit vectors $P$ (**Fig 2b**), whose cardinality is bounded by **Lemma 2**. It projects $X$ onto $P$ and iteratively estimates singular values and the null space of the approximated left singular vectors (**Fig 2c**).

---

**Algorithm 1: Singular Value Approximation** (Based on Random Projection)

---

**Inputs:** $X^{R \times R}$, $N^R$ randomly generated unit vectors denoted as $P \in \mathbb{R}^{R \times N^R}$, cutoff $\theta$
**Outputs:** Estimated singular values $\sigma_1, \sigma_2, ..., \sigma_R$
**Singular Value Approximation**$(X, P, \theta)$:
$Y \leftarrow XP$
Generate vector $E$, $E_j \leftarrow \sqrt{\sum_{i=1}^{R} Y_{ij}^2}$
$\sigma_1 \leftarrow \max \{E_j | 1 \le j \le N^R\}$
$\sigma_R \leftarrow \min \{E_j | 1 \le j \le N^R\}$
$P_{(1)} = \underset{P_{\cdot, j}}{\arg \max} E$, $\mathcal{P}_0^{nullsp} \leftarrow \{P_j | P_j \text{ are columns of } P\}$
**for** *r in 1,...,R − 1* **do**
$\quad | \quad \mathcal{P}_r^{nullsp} \leftarrow \{P_j | P_j \in \mathcal{P}_{r-1}^{nullsp}, \max\{\cos(P_j, P_{(1)}), ..., \cos(P_j, P_{(r)})\} < \cos(\theta)\}$
$\quad | \quad \sigma_{r+1} \leftarrow \max \{E_j | \text{the corresponding } P_j \text{ of } E_j \in \mathcal{P}_r^{nullsp}\}$
$\quad | \quad P_{(r+1)} \leftarrow \underset{P_j}{\arg \max} \{E | P_j \in \mathcal{P}_r^{nullsp}\}$
**end**
**return** $\{\sigma_1, \sigma_2, ..., \sigma_R\}$

---

In **Algorithm 1**, $P_{(r)}$ and $\mathcal{P}_r^{nullsp}$ are estimated $r$th left singular vector and the null space of the linear space spanned by the first $r$ left singular vectors, respectively. As randomly generated vectors cannot be stringently orthogonal, $\theta$ is a hyper-parameter that determines the randomly generated vectors that are in the null space of $P_{(r)}$. Noted, the null space of the linear space spanned by each $P_j$ does not rely on $X$ that can be computed before the random projection. Thus, the random projection and iterative computing of $\sigma_r$ and $P_{(r)}$ only involve inner product, max, and sum of squares, which can be efficiently and parallelly computed on GPU for a very large set of small matrices.

We have evaluated the computational efficiency and accuracy of **Algorithm 1** versus conventional QR decomposition-based SVD on both GPU and CPU servers (see more details in APPENDIX). We tested the two methods 50 times on $10^5$, $10^6$, $10^7$ and $10^8$ $2 \times 2$, $10^5$ $4 \times 4$, and $10^5$ $8 \times 8$ matrices. The averaged normalized root mean squared error between estimated and true singular values is 0.07. We observed that **Algorithm 1** used $10^{-5} - 10^{-2}$ seconds, which is consistently about $10^5$ faster than QR decomposition-based SVD. The max pooling step can be further optimized by first clustering the random unit vectors into groups of high cosine similarities (**Fig 2d**), as detailed in APPENDIX.

## 3.2 THE RPSP FRAMEWORK

**Algorithm 2** and **Fig 3** illustrate the main framework of RPSP. The inputs of RPSP include a matrix $X^{M \times N}$ and hyper-parameters. The output are identified local low rank matrices, denoted as $\{X_{I_k \times J_k}\}, k = 1, ..., K$. The initialization step of RPSP generates random unit vectors for estimating singular values of small matrices. Specifically, $N_t$ random vectors of length $2^t$, denoted as $P_t, t = 1...T$ will be generated, where $T$ is the number of layers for submatrices propagation.

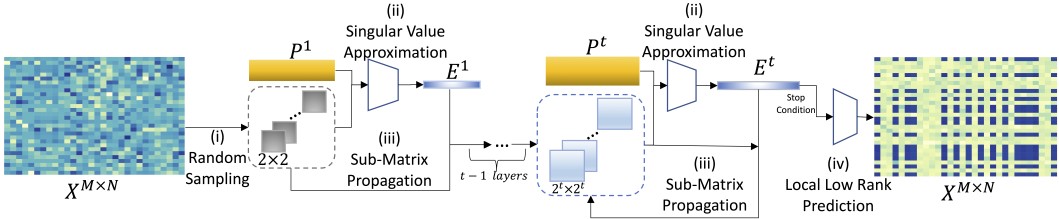

Figure 3: The framework of the RPSP algorithm.

RPSP first randomly samples $L_1$ $2 \times 2$ submatrices from $X$, whose singular values are estimated by **Algorithm 1** against $P_1$, as described in 3.1. Noted, $2 \times 2$ is the smallest unit submatrix that possess a low rank structure. Our evaluation of **Algorithm 1** suggests that the computation for $L^1 = 10^1 12 \times 2$ matrices uses less than 10s. The value $\frac{\sigma_1}{||P||_*}$ characterizes the low rank property of a $2 \times 2$ matrix $P$, where $\sigma_1$ and $||P||_*$ denote the first singular value and nuclear norm of $P$. The low rank property of the $2 \times 2$ submatrices is further propagated by a weighted sampling to generate $L_2$ $4 \times 4$ submatrices. Specifically, a pair of none overlapped $2 \times 2$ matrices were randomly sampled by a probability weighted by the average of their $\frac{\sigma_1}{||P||_*}$ values. The row and column induces of the two samples matrices form a new $4 \times 4$ submatrix. The singular values of the $4 \times 4$ submatrices will be estimated by random projection and orthogonal pooling against $P_2$. This procedure will be iteratively conducted $T$ times, by which $L_T$ $2^T \times 2^T$ submatrices will be randomly sampled weighted by the low rankness propagated through the $T$ layers, whose singular values will be estimated. For each $t = 1, ..., T$, RPSP also computes a $M \times N$ scoring matrix $S^t$, in which $S^t_{ij}$ stores the frequency of observing a large value of $\frac{\sigma_1}{||P||_*}$ among all the sampled $2^t \times 2^t$ submatrices that hits to $X_{ij}$. $S^T_{ij}$ can be viewed as an approximation of the probability that $X_{ij}$ is contained by a LLR submatrix with a size of $2^T \times 2^T$ or larger. The local low rank submatrices in $X$ can be further identified by a co-clustering over $S^T$. RPSP (**Algorithm 2**) consists the following sub algorithms:
(i) **Singular Value Approximation** (**Algorithm 1**) computes singular values for the $2^t \times 2^t$ submatrices, as described in 3.1.
(ii) **submatrix Propagation** generate $2^{t+1} \times 2^{t+1}$ submatrices by randomly sample pairs of non-overlapped $2^t \times 2^t$ submatrices with a probability weighted by the average of the low rankness score. This approach enable the propagation of the low rankness of two small submatrices to a larger one if the two small submatrices truly hit one local low rank submatrix. (Detailed in APPENDIX)
(iii) **Local Low Rank Prediction** reconstructs the local low rank matrices in $X$ based on the scoring matrix $S^T$ (Detailed in APPENDIX).

In the **Algorithm 2**, $P_t$ denote the sets of randomly generated unit vectors; $\mathcal{R}_t$ denote the sets of the $2^t \times 2^t$ submatrices randomly sampled ($t = 1$) or weighted sampled ($t = 2, ..., T$) by **submatrix Propagation**; $\mathcal{R}_t[j]$ denotes the $j$th submatrix in $\mathcal{R}$; $E^{t,t \times L_t}$ store the estimated singular values; $LowRankScore^T$ is a vector stores the top singular value divided by the nuclear norm of each $2^T \times 2^T$ submatrix and $S^T$ denotes the scoring matrix, where $S^T_{ij}$ is the frequency of observing $LowRankScore^T > C$ for all the submatrices that contains $X_{ij}$. Noted, the hyper-parameters $T$ and $L_t$ can be easily determined based on the computational capability while $C$ and $N_t$ can be determined based on the level of errors can be tolerated (see details in APPENDIX).

## 4 EXPERIMENTS ON SYNTHETIC DATA

We evaluated the overall performance and computational cost of RPSP on different scenarios of MLLRR problem compared with SOTA methods on a comprehensive setup of synthetic datasets.

### 4.1 EXPERIMENTAL SETUP

We simulate $X \in \mathbb{R}^{M \times N}$ as $X = \sum_{k=1}^{K} X^k + E$. Here, entries in $X^k$ is padded by zero except for those indexed by $I_k$ and $J_k$, corresponding to a local low rank submatrix with $m_k$ rows and $n_k$ columns. $E$ is background noise simulated by $E_{i,j} \sim N(0, \alpha_k * sd), \forall i \in I_k, j \in J_k$; and the rest of the entries in $E$ follows $N(0, sd)$. In evaluating the algorithm's scalability, we allow $M$ and $N$ to have three different values. Otherwise, we let $M = N = 1000$. To simulate $X^k_{I_k \times J_k}$, we first simulated $Y_k$ as $Y_k = U_k V_k^T$, where $U_k \in \mathbb{R}^{m_k \times r_k}$ and $V_k \in \mathbb{R}^{n_k \times r_k}$, and entries in $U_k, V_k$ all follow $U(0, 1)$. Then $X^k_{I_k, J_k}$ is simulated as $Y_k - \overline{Y_k} + \mu_k$. Here, $\overline{Y_k}$ denotes the element-wise mean

---

**Algorithm 2: RPSP**

---

**Inputs:** $X^{M \times N}$, hyper-parameters $T, N_t, L_t, C, t = 1, ..., T$
**Outputs:** The indices set $\{\mathcal{I} \times \mathcal{J}\}$, where $I_k \in \mathcal{I}, J_k \in \mathcal{J}$, $X_{I_k \times J_k}$ is a local low rank matrix.
**RPSP**$(X, T, N_t, L_t, C)$:
**for** *t in 1,...,T* **do**
  | $P_t \leftarrow \{N_t$ randomly generated unit vectors of length $2^t\}$
**end**
$\mathcal{R}_1 \leftarrow \{L_1 \ 2 \times 2$ submatrices randomly sampled from $X\}$
$E^{1,2 \times L_1} \leftarrow$ **Singular Value Approximation**$(\mathcal{R}_1, P_1)$
**for** *t in 2,...,T* **do**
  | $\mathcal{R}_t \leftarrow$ **submatrix Propagation**$(X, \mathcal{R}_{t-1}, E^{t-1}, L_t)$
  | $E^{t,2^t \times L_t} \leftarrow$ **Singular Value Approximation**$(\mathcal{R}_t, P_t)$
**end**
**for** *j in 1,...,$L_T$* **do**
  | $LowRankScore^T[j] \leftarrow \frac{E^T_{1,j}}{\sum^T_{i=1} E^T_{i,j}}$
**end**
**for** *i in 1,...,M* **do**
  | **for** *j in 1,...,N* **do**
  |   | $S^T_{ij} \leftarrow$ frequency of $LowRankScore^T[k] > C$ for all $\mathcal{R}_t[k]$ contains $X_{ij}$
  | **end**
**end**
$\mathcal{I} \times \mathcal{J} \leftarrow$ **Local Low Rank Prediction**$(S^T)$
**return** $\mathcal{I} \times \mathcal{J}$

---

of matrix $Y_k$, and hence $\mu_k$ mimics the overall mean of the $k$-th pattern matrix, and $\alpha_k$ mimics the relative noise level of the local low rank matrix to the overall background noise matrix. In total, we obtained 284 different simulation scenarios, each has 5 repetitions, which include:

(1) **Perturbed pattern mean:** pattern mean $\mu_k = \beta_k * sd$, where $\beta_k$ is a sequence from 0 to 3 with step size 0.1; relative noise level $\alpha_k = \{0, 0.1\}$; pattern size $m_k = n_k = \{200, 500\}$.
(2) **Perturbed background error:** pattern mean $\mu_k = \beta_k * sd$, where $\beta_k = \{0, 0.1\}$; relative noise level $\alpha_k$ is a sequence from 0 to 3 with step size 0.1; pattern size $m_k = n_k = \{200, 500\}$.
(3) **Perturbed pattern size:** pattern mean $\mu_k = \beta_k * sd$, where $\beta_k = \{0, 0.1\}$; relative noise level $\alpha_k = \{0, 0.1\}$; pattern size $m_k = n_k$ is a sequence from 100 to 500 with step size of 20.

We evaluated the method performance of RPSP and selected SOTA methods on these synthetic datasets, based on how well the identified patterns hit the true ones, and avoid the background noise. We label the entries hitting true patterns as "positive" and the rest as "negative", and the True Positive (TP), False Negative (FP), False Positive (FP), and True Negative (TN) occurrences are defined as the number of "positive" entries that are identified as pattern (TP) or background (FP), or the number of "negative" entries that are identified as pattern (FP) or background (TN). The overall prediction accuracy is defined as $\frac{TP+TN}{TP+TN+FP+FN}$.

### 4.2 PERFORMANCE EVALUATION OF RPSP

We benchmarked RPSP with five SOTA methods, namely Bregman co-clustering (CC) Banerjee et al. (2007) and Plaid Lazzeroni & Owen (2002), two sparse matrix decomposition methods (SSVD Yang et al. (2014) and SPCA Zou et al. (2006)), and one anchor based method LLORMA Lee et al. (2016). Detailed parameter setting of RPSP and other methods are provided in APPENDIX.

**Accuracy in solving the LCV and LLR problems. Fig 4a-c** illustrated the accuracy ($y$-axis) of RPSP (red) and other methods for solving the LCV and LLR problem in different scenarios. Overall, RPSP achieved higher than 0.8 accuracy under most settings, which is consistently higher than baseline methods. On the dense data, SPCA failed to identify any pattern while CC detects the whole matrix as one pattern, hence these two methods were excluded from further analysis. RPSP is the only method that can identify local low rank pattern when its mean is close to the background mean (**Fig 4a**), i.e., the LRR problem without the LCV property. As expected, we observed the prediction

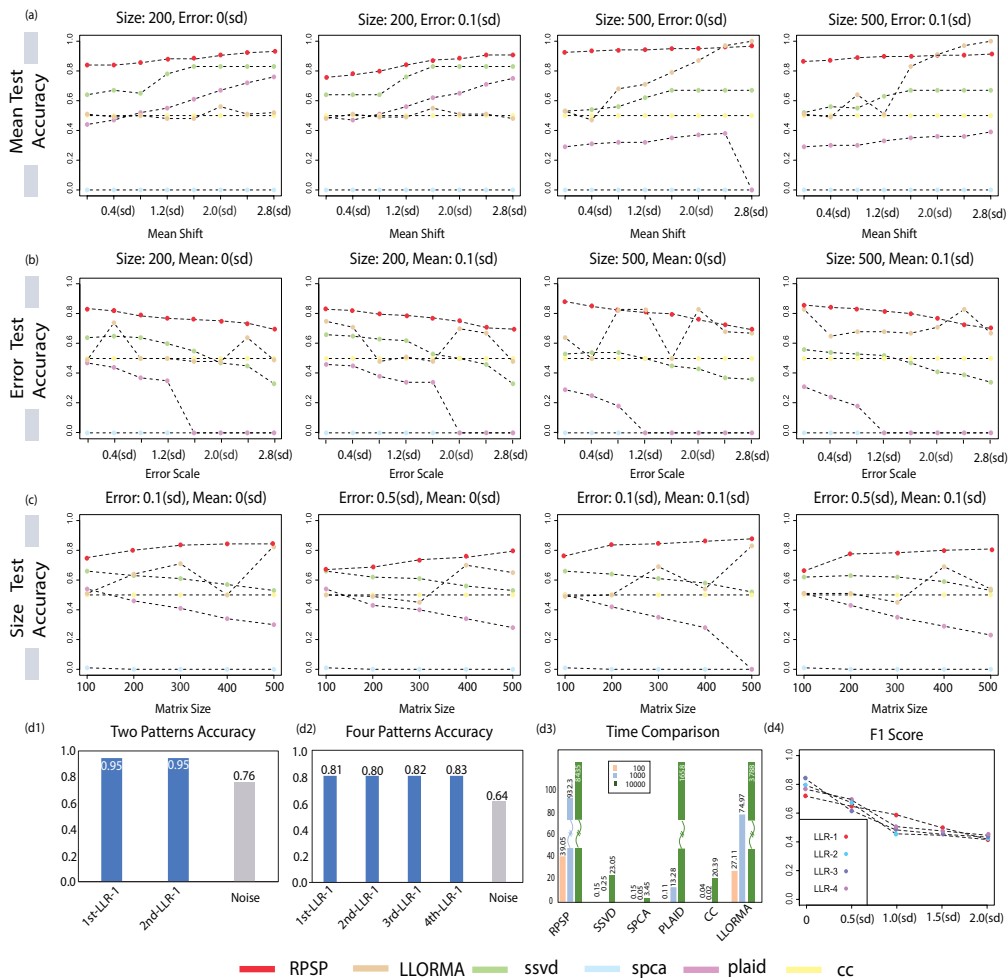

Figure 4: Benchmark of RPSP on Synthetic Data.

accuracy of the baseline methods to increase as the mean difference becomes larger (**Fig 4a**), and all methods to have decreased performance with the increase of the noise level (**Fig 4b**). RPSP and LLORMA are more robust to high noise level compared to SSVD and Plaid. The size test suggested that RPSP can accurately identify the pattern when its size is even smaller than $100 \times 100$ in a $1000 \times 1000$ matrix (**Fig 4c**). When the pattern size increases, the prediction accuracy of RPSP and LLORMA also increase, but not SSVD or Plaid. An explanation is that a larger pattern is easier to be hit by the randomly sampled submatrices in RPSP or the anchoring in LLORMA, while SSVD and Plaid rely on the pattern sparsity assumption and are less sensitive to large patterns.

**Power in detecting multiple patterns and the submatrices of different ranks.** Where there exists more than 1 local rank-1 sub matrices, RPSP again achieved high performance (**Fig 4d1-2**). The way RPSP detects local low rank matrices is from the scoring matrices, which is less impacted by the number of patterns. It is noteworthy that we focus on the LLR problem in a dense matrix, while LLORMA and Plaid are more efficient on the LCV problem in a sparse matrix. On the dense matrix, all the baseline methods failed to identify the LLR pattern when the mean difference of the pattern and the background is low. We also evaluated RPSP on identifying LLR patterns of different dimensions (**Fig 3d4**). Our results demonstrated that RPSP has a high robustness in detecting patterns of different dimensions. Noted, the specificity of RPSP is always bounded by $1 - a$, where $a$ is the probability of presence of a local low rank matrix in a noise matrix.

**Running time.** We evaluated the time consumption of the methods on dense matrices of three sizes, $M = N = 10^2, 10^3, 10^4$ (**Fig 3d3**). The running time of RPSP, Plaid and LLORMA are at a similar level. Detailed experimental results and parameters of the GPU and CPU server for the experiment are provided in APPENDIX.

Table 1: Experiment on real-world data.

| | MovieLens | | | | | | | | GSE103322 | | | | | | | |
| | Low Rankness | | | Size | | | CR | Time(s) | Low Rankness | | | Size | | | CR | Time(s) |
|---|---|---|---|---|---|---|---|---|---|---|---|---|---|---|---|---|
| RPSP | **0.82** | **0.8** | **0.76** | **1.90E+03** | **2.50E+03** | **2.50E+03** | 0.1 | 52.03 | **0.7** | **0.5** | **0.47** | **1.10E+05** | **1.00E+05** | **8.90E+04** | **0.08** | 300.6 |
| | 0.76 | 0.73 | 0.72 | 1.70E+03 | 7.00E+02 | 7.00E+02 | | | 0.44 | 0.43 | 0.43 | 6.00E+04 | 4.90E+04 | 4.00E+04 | | |
| LLORMA | 0.58 | 0.54 | 0.52 | 2.50E+03 | 2.40E+03 | 2.50E+03 | 0.06 | 41.96 | 0.39 | 0.39 | 0.38 | 1.90E+04 | 1.90E+04 | 1.90E+04 | 0.02 | 637 |
| | 0.52 | 0.52 | 0.51 | 2.50E+03 | 2.30E+03 | 2.50E+03 | | | 0.24 | 0.21 | 0.19 | 1.90E+04 | 1.90E+04 | 1.90E+04 | | |
| SSVD | 0.59 | 0.5 | 0.5 | 2.50E+03 | 2.60E+03 | 2.30E+03 | 0.06 | 1.37 | 0.41 | 0.29 | 0.25 | 1.90E+04 | 1.90E+04 | 1.90E+04 | 0.02 | 49.59 |
| | 0.5 | 0.5 | 0.49 | 2.30E+03 | 2.50E+03 | 2.50E+03 | | | 0.24 | 0.2 | 0.19 | 1.90E+04 | 1.90E+04 | 1.90E+04 | | |
| SPCA | 0.48 | 0.48 | 0.48 | 2.30E+03 | 2.30E+03 | 2.30E+03 | 0.06 | 0.15 | 0 | 0 | 0 | 198 | 198 | 198 | 2.00E-04 | 3.55 |
| | 0.48 | 0.48 | 0.48 | 2.50E+03 | 2.50E+03 | 2.50E+03 | | | 0 | 0 | 0 | 198 | 198 | 198 | | |
| PLAID | NA | NA | NA | NA | NA | NA | NA | NA | 0.31 | NA | NA | 8.10E+03 | NA | NA | 1.00E-03 | 182.84 |
| | NA | NA | NA | NA | NA | NA | | | NA | NA | NA | NA | NA | NA | | |
| CC | 0.51 | NA | NA | 6.00E+04 | NA | NA | **0.18** | 363 | 0.28 | NA | NA | 4.10E+05 | NA | NA | 0.05 | 951 |
| | NA | NA | NA | NA | NA | NA | | | NA | NA | NA | NA | NA | NA | | |

## 5 EXPERIMENTS ON REAL-WORLD DATA

We benchmarked RPSP on two real-world datasets of different density rates (proportion of non-zero entities in the overall input matrix), error distributions, and local low rank patterns, namely (1) the MovieLens data and (2) single cell RNA-seq data. Details of data processing and algorithm settings are given in APPENDIX. Four metrics were utilized to evaluate the performance of each method, namely (1) the Low Rankness, calculated as the averaged $\frac{\sigma_1}{||P||_*}$ and (2) the averaged Size of the identified submatrices $P$, (3) the total Coverage Rate(CR), defined as the total number of entries in the top-$k$ patterns divided by the size of the input matrix, and (4) the Running Time. In addition, context specific meaning was evaluated based on prior knowledge.

**Application to MovieLens data**. MovieLens 25M data is a common benchmark dataset. It contains the ratings of 62,000 movies by 162,000 users provided by GroupLensHarper & Konstan (2015). We selected the top 600 active users(rows) and the 600 most rated movies(columns), reaching a density rate of $20.0\%$. This is a relatively sparse dataset. Indeed, on sparse data, all baseline methods tend to identify LCV submatrices, but still RPSP is more favorable than others in terms of the low rankness, coverage rate. As shown in **Table 1**, the low rankness of the top six significant local low rank matrices identified by RPSP is consistently higher ($\sim0.75$) than the ones detected by baseline methods ($\sim0.5$). While the patterns detected by RPSP is of smaller sizes, their coverage rate is higher than LLORMA, SSVD and SPCA because RPSP tends to detect non-overlapped patterns while the top patterns identified by baseline methods are heavily overlapped.

**Application to single cell (sc)RNA-seq data**. The local low rank submatrices in scRNA-seq correspond to sub-population of cells with distinct functions and the error distribution is heteroscedastic and non-Gaussian Hou et al. (2020); Wan et al. (2019a). We applied RPSP and baseline methods on a head and neck cancer scRNA-seq data, namely GSE103322 Puram et al. (2017). Here we removed certain genes and cells with low expression, bringing the density rates to $75.71\%$. As shown in **Table 1**, the low rankness, size and coverage rate of the top patterns identified by RPSP are consistently higher than the ones detected by baseline methods. We conducted enrichment analysis of the gene features in each detected submatrix to examine their interpretability Huang et al. (2007). We found three local low rank matrices identified by RPSP that significantly enrich biological functions including cell metabolism, cell proliferation, and antigen presentation.

In summary, RPSP outperforms all baseline methods in detecting local low rank matrices on the real-world datasets, in terms of the low rankness, size, coverage rate, and contextual interpretability of detected local low rank patterns. The running time of RPSP is at a similar level of baseline methods.

## 6 CONCLUSION

In this work, we provided a new computational framework, namely RPSP, to detect local low rank matrices. RPSP is supported by rigorously derived mathematical theories. While existing methods mainly focused on one sub-problem of MLLRR, our developed RPSP is the first method capable of handling the general MLLRM problem. RPSP utilizes an efficient random projection and GPU-based computation of singular values for a large set of small matrices. It propagates the low rankness identified from small matrices to larger ones to identify local low rank submatrices of coherent patterns. On both synthetic and real-world experiments, we demonstrated that RPSP outperforms all baseline methods on the LLR problems for data of different sparsity level and error distributions. Particularly, RPSP could detect low rank submatrices even when its mean structure is not distinguishable from the background, or when the error distribution is heteroscedastic.

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

# A APPENDIX

## A.1 SUB-ALGORITHMS OF RPSP

### A.1.1 SUBMATRIX PROPAGATION

The **Algorithm 3 submatrix Propagation** conducts weighted sampling to generate $2^t \times 2^t$ submatrices based on the low rankness of $2^{t-1} \times 2^{t-1}$ submatrices. The input of **Algorithm 3 submatrix Propagation** include the input matrix $X$, sampled $2^{t-1} \times 2^{t-1}$ submatrices, their singular values $E^t$, and the number of to be generated $2^t \times 2^t$ submatrices $L_t$.

---

**Algorithm 3: submatrix Propagation**

---

**Inputs:** $X, \mathcal{R}_{t-1}, E^{t-1}, L_t$
**Outputs:** $\mathcal{R}_t$
**submatrix Propagation**$(X, \mathcal{R}_{t-1}, E^{t-1}, L_t)$:
$\mathcal{R}_t \leftarrow \varnothing$
**while** $|R_t| < L_t$ **do**
    $Prob \sim U(0,1)$
    Randomly pick two non-overlapping submatrices from $\mathcal{R}_{t-1}$, denoted as $\mathcal{R}_{t-1}[i]$ and
      $\mathcal{R}_{t-1}[j]$
    $W \leftarrow \dfrac{E_{1,i}^{t-1} E_{1,j}^{t-1}}{\sum_{k=1}^{2^{t-1}} E_{k,i}^{t-1} \sum_{k=1}^{2^{t-1}} E_{k,j}^{t-1}}$
    **if** *Prob<W* **then**
        $I \leftarrow$ Row indices of $\mathcal{R}_{t-1}[i]$ and $\mathcal{R}_{t-1}[j]$
        $J \leftarrow$ Column indices of $\mathcal{R}_{t-1}[i]$ and $\mathcal{R}_{t-1}[j]$
        $append(\mathcal{R}_t, X_{I \times J})$
**end**
**return** $\mathcal{R}_t$

---

### A.1.2 LOCAL LOW RANK PREDICTION

RPSP further utilizes **Algorithm 4 Local Low Rank Prediction** to identify and reconstruct the local low rank matrices. RPSP computes a $M \times N$ scoring matrix $S^T$, in which $S_{ij}^T$ stores the frequency of observing a large value of $\frac{\sigma_1}{||P||_*}$ among all the sampled $2^T \times 2^T$ submatrices that hits to $X_{ij}$. Hence, $S_{ij}^T$ can be viewed as an approximation of the probability that $X_{ij}$ is contained by a LLR submatrix with a size of $2^T \times 2^T$ or larger. Here we applied the **Spectral Co-Clustering** method developed by Dhillon et al Dhillon (2001) and the python library provided by scikit-learn Pedregosa et al. (2011) on $S^T$ to identify local low rank submatrices. With the indices of each possible local low rank matrix identified, the local patterns were ranked by the level of their top singular values normalized by the sum of all singular values. Here the local patterns of the top K significant low rank property or with the top singular values large than a certain threshold form the final output of RPSP.

---

**Algorithm 4: Local Low Rank Prediction**

---

**Inputs:** $S^{T,M \times N}$
**Outputs:** $\mathcal{I} \times \mathcal{J}$
**Local Low Rank Prediction**$(S^{T,M \times N})$:
$\mathcal{I} \times \mathcal{J} \leftarrow$ **Spectral Co-Clustering**$(S^T)$
**return** $\mathcal{I} \times \mathcal{J}$

---

### A.1.3 OPTIMIZE THE MAX POOLING WITH RESPECT TO NULL SPACE IN **ALGORITHM 1**

The max pooling with respect to null space can be further optimized by clustering the random unit vectors into groups of high cosine similarities. Specifically, the randomly sampled unit vectors were first clustered by using K-mean of their cosine distance (1-cosine similarity). Then after $P_{(1)}, ..., P_{(r)}$ were identified, the null space of the linear span of $\{P_{(1)}, ..., P_{(r)}\}$ was estimated by the union of the clusters whose center $P^C$ satisfies $\max\{\cos(P^C, P_{(1)}), ..., \cos(P^C, P_{(r)})\} < \cos(\theta)\}$. This approach effectively reduce the number of cosine distance needed to be computed.

### A.1.4 ASSESSMENT OF HYPER-PARAMETERS OF RPSP

RPSP has four hyper-parameters $T$, $C$, $L_t$, and $N_t$. $L_t$ (number of randomly sampled or propagated submatrices) can be determined based on the input matrix size and computational capacity. $T$ (number of layers for submatrix propagation) is set as 4 for efficient computation. $C$ (threshold of $LowRankScore$) can be computed by randomly sampling $2^T \times 2^T$ submatrices of pure noise from randomly shuffled $X$ and generating an empirical null distribution of $LowRankScore$. $N_t$ (number of random unit vectors) can be determined by **Lemma 2**.

## A.2 MATHEMATICAL DERIVATIONS AND CONSIDERATIONS

### A.2.1 MATHEMATICAL CONSIDERATIONS OF THE MLLRR PROBLEM

The biggest challenge with local low rank submatrix detection lies in that neither the row or column indices of the submatrix is known. As given in Definition 2, the low rankness property of a submatrix is evaluated through the computation of its singular values, which apparently can't be evaluated until the submatrix has been presented. However, it is computationally impossible to go through all the submatrices of an input matrix. RPSP grows a submatrix of low rank from smaller ones, which utilizes two facts. Firstly, for a low rank matrix $X^{M \times N}$ of rank $r \ll min(M, N)$, the self consistency property suggests that any $M_0 \times N_0$ submatrix ($M_0, N_0 \geq r$) randomly sample from $X$ is most likely to have a rank of $r$ (Lemma 1 in Owen et al. (2009)). Secondly, for a given matrix, the total number of square submatrices of dimension $M_0$ grows exponentially with $M_0$. The first fact indicates that, any submatrix of low rank is a collage or complete coverage of its own (smaller) submatrices, which are also of low rank. The second fact indicates that the only way for us to grow a local low rank submatrix is to start from the much smaller submatrices. In fact, for $M_0$ as small as 2, it is computationally feasible for us to obtain a full collection of $M_0 \times M_0$ submatrices that could densely cover $X^{M \times N}$. By teasing out all the $M_0 \times M_0$ submatrices of low rank, we could then gradually build them up into larger low rank submatrices. The evaluation of the low rankness for a large number of submatrices now becomes computationally expensive. In RPSP, our biggest contribution is that we have developed a singular value approximation method using random project to efficiently evaluate the low rankness of any given submatrix, making it possible for us to build a submatrix from its parts.

A few examples can illustrate that why a global search cannot effectively solve the MLLRR problem. We consider the following square matrices:

(1) $X^M \times M$ in which $X_{ij} \sim N(0, 1)$ $i.i.d.$. Here $X$ is a matrix of standard Gaussian error. The largest singular value of $X$ is about $2\sqrt{M}$.
(2) $Y^M \times M$ in which $Y_{i,\cdot} \equiv Y'$, $Y'[i] \sim N(0, 1)$ $i.i.d.$. Here $Y$ is a matrix of rank=1 that have the same level of mean and standard deviation as $X$. Noted, the largest singular value of $Y$ is about $M$.
(3) $Z^{M \times M}$ in which $Z_{ij} \equiv a$. The largest singular value of $Y$ is $M \times a$.

Hence for a low rank sub-matrix of size $\sqrt{2M} \times \sqrt{2M}$ or smaller, whose mean and standard deviation is not different to the background's, it is less likely to be identified by a global SVD as the largest singular value of the sub-matrix is about the same level of the largest singular value of the background noise matrix. However, if the low rank sub-matrix has a spiked mean, its largest singular valued will be amplified by the spiked mean and the top singular vector of the whole matrix is naturally sparse. Hence, a LCV problem is more likely to be solved by a global search while the LLR problem of insignificant mean difference between pattern and background is less likely to be detected by a global search. So, it is necessary to think alternative approach to solve the general MLLRR problem. Noted,

the idea of screening a large set of small submatrices and propagate the low rank property of smaller ones to bigger submatrices only involves the computing of singular values of local patterns. Hence, we do not expect that the RPSP method may have disparate performances in solving LCv and LRR problems.

### A.2.2 MATHEMATICAL FORMULATIONS OF SOTA METHODS

Co-clustering methods simultaneously clusters rows and columns of a two-dimensional data matrix. The general assumption is that the targeted submatrix has a larger or small mean value comparing to the background noise. The Bregman co-clustering method generates a matrix partition $I_k, J_k$ by preserving the maximum information of data $X$ within the partitions. The approximation error $M(I, J) - M(\tilde{I}, \tilde{J})$ represents the difference of the preserved information and original data, here $M(I, J)$ is the mutual information and $M(I, J) - M(\tilde{I} - \tilde{J}) = KL(dist_1(I, J)||dist_2(I, J))$. Laura et al. proposed the Plaid model to detect the submatrix by fitting each entry $X_{ij}$ with $K$ layers and make sure the summation of all layers $\sum_{k=1}^{K} \mu_k I_k J_k$ approximate the original value. Sparse SVD based methods identify local low rank matrix by adding L1 sparse panelty to a global truncated SVD fitting. However, this type of methods still demands distinct mean difference between pattern and background and trend to detect large low rank pattern that may explain the variance of the whole matrix. Lee et al. proposed the LLORMA method by using prior knowledge to select anchors of local low rank patterns. As listed in table 1, $K_\Omega^h$ is the kernel function with bandwith $h$ to smooth the projection value $P_\Omega(\cdot)$ near the anchor points $\Omega$. However, this type of methods,highly depends on prior knowledge that cannot solve the general MLLRR problem.

Table 2: Existing methods of MLLRR

| Methods | Examples | Formulation | Tasks | Assumption |
|---|---|---|---|---|
| Co-clustering | Bregman; Plaid | $\min_{I_k, J_k, \mu_k} \sum_k \sum_{i \in I_k, j \in J_k} d(x_{ij}, \mu_k)$ | LCV | Matrix partition |
| Matrix decomposition | SSVD; SPCA | $\min_{U,V}(||X - UV^T||_F^2 + \lambda_u||U||_1 + \lambda_v||V||_1)$, | LCV LLR | Sparse patterns |
| Anchor based methods | LLORMA WEMAREC | $\min_{\hat{I}, \hat{J}, \hat{X}} (K_{X[\hat{I}, \hat{J}]} \odot P_{X[\hat{I}, \hat{J}]}(X - \hat{X}))$ | LLR | submatrix detection |

### A.2.3 PROOFS OF LEMMA 1

**Lemma 1.** For a given dimension $R$, denote $X^{R \times R}$ as an input matrix and $P \in \mathbb{R}^{R \times N^R}$ as a matrix of $N^R$ randomly generated unit vectors in $\mathbb{R}^R$. $Y = XP$ denotes a random projection of $X$, then $\lim_{N^R \to \infty} \max_{1 \le j \le N^R} \sqrt{\sum_{i=1}^{R} Y_{ij}^2} = \sigma_1$ and $\lim_{N^R \to \infty} \min_{1 \le j \le N^R} \sqrt{\sum_{i=1}^{R} Y_{ij}^2} = \sigma_R$, here $\sigma_1$ and $\sigma_R$ are the largest and smallest singular values of $X$. Denote $P_{(1)} = \arg\max_{P_{\cdot,j}} \sqrt{\sum_{i=1}^{R} Y_{ij}^2}$, $\lim_{N^R \to \infty} \max_{P_{\cdot,j} \in Sp(P_{(1)})^\perp} \sqrt{\sum_{i=1}^{R} Y_{ij}^2} = \sigma_2$, where $Sp(P_{(1)})^\perp$ denotes the null (or complemented) space of the linear space spanned by $P_{(1)}$. Similarly, define $P_{(r)} = \arg\max_{P_{\cdot,j} \in Sp(P_{(1)},...,P_{(r-1)})^\perp} \sqrt{\sum_{i=1}^{R} Y_{ij}^2}$, $\lim_{N^R \to \infty} \max_{P_{\cdot,j} \in Sp(P_{(1)},...,P_{(r-1)})^\perp} \sqrt{\sum_{i=1}^{R} Y_{ij}^2} = \sigma_r$, for $r \in 3, ..., R-1$.

*Proof.* Noted, $\sum_{i=1}^{R} Y_{ij}^2$ is the norm of the projection of $X$ onto $Y_{\cdot,j}$. $Y_{\cdot,j} = XP_{\cdot,j} = U\Sigma V^T P_{\cdot,j} = \sum_{k=1}^{R} U_{\cdot,k} \Sigma_{kk} V_{k,\cdot}^T P_{\cdot,j}$, here $U\Sigma V^T$ is the SVD of $X$. Then we have $\sum_{i=1}^{R} Y_{ij}^2 = \sum_{i=1}^{R} (\sum_{k=1}^{R} U_{ik} \Sigma_{kk} V_{k,\cdot}^T P_{\cdot,j})^2 = \sum_{k=1}^{R} \sum_{i=1}^{R} (U_{ik})^2 (\Sigma_{kk})^2 (V_{k,\cdot}^T P_{\cdot,j})^2 = \sum_{k=1}^{R} (\Sigma_{kk})^2 (V_{k,\cdot}^T P_{\cdot,j})^2$ as $U$ is orthogonal, both $\Sigma_{kk}$ and $V_{k,\cdot}^T P_{\cdot,j}$ are scalars. Hence the largest and smallest $\sum_{i=1}^{R} Y_{ij}^2$ is $\Sigma_{11}$ and $\Sigma_{RR}$, which are achieved when $P_{\cdot,j}$ is $V_{1,\cdot}^T$ and $V_{R,\cdot}^T$, respectively. As $Sp(Y_{(1)}, ..., Y_{(r-1)})^\perp$ is the null space of the linear span of $Y_{(1)}, ..., Y_{(r-1)}$, the largest projection of $X$ onto this space is $\sigma_r$ when $P_{\cdot,j}$ is $V_{r,\cdot}^T$. $\square$

## A.3 EXPERIMENTAL DETAILS

Detailed experimental parameters, data and analysis are provided below. We conducted experiments on a GPU server of Cray HPE EX architecture featured with 64 nodes, 4 A100 GPUs, 64 core and 256GB per node and a CPU server features 640 compute nodes, each equipped with 256 GB of memory and two 64-core, 2.25 GHz, 225-watt AMD EPYC 7742 processors.

Table 3: The Singular Value Approximation and SVD Running Time C.

| q=2,nc=5 | Inner time(gpu) | Inner time(cpu) | Svd time(cpu) |
|---|---|---|---|
| ns=1e4 | 0.00002 | 0.0003 | 0.9 |
| ns=1e5 | 0.00002 | 0.0031 | 0.1166 |
| ns=1e6 | 0.000036 | 0.0328 | 1.1563 |
| ns=1e7 | 0.0002 | 0.3664 | 11.6114 |
| ns=1e8 | 0.037 | 3.6 | 116 |
| q=4,nc=27 | Inner time(gpu) | Inner time(cpu) | Svd time(cpu) |
| ns=1e4 | 0.000027 | 0.00235 | 0.0325 |
| ns=1e5 | 0.000028 | 0.02436 | 0.3077 |
| q=8,nc=767 | Inner time(gpu) | Inner time(cpu) | Svd time(cpu) |
| ns=1e4 | 0.000032 | 0.0959 | 0.0828 |
| ns=1e5 | 0.000059 | 0.9422 | 0.8171 |
| q=16,nc=588804 | Inner time(gpu) | Inner time(cpu) | Svd time(cpu) |
| ns=500 | 0.000349 | NA | NA |
| q=16,nc=3544 | Inner time(gpu) | Inner time(cpu) | Svd time(cpu) |
| ns=1000 | 0.00003 | 0.1071 | 0.0268 |
| ns=10000 | 0.000056 | 1.07 | 0.2542 |

### A.3.1 BENCHMARK OF **ALGORITHM 1: SINGULAR VALUE APPROXIMATION**

We have evaluated the computational efficiency and accuracy of versus on conventional QR decomposition-based computation of singular values. We tested **Algorithm 1** on the GPU server and QR decomposition-based SVD on both of the GPU server and a CPU server. Noted, due to the nature of QR decomposition, it is slower on GPU compared to CPU. We tested the two methods 50 times on $10^6$ $2 \times 2$, $10^5$ $4 \times 4$, and $10^5$ $8 \times 8$ matrices. The **Algorithm 1** used less than $10^{-4}$ second, which is on average about $10^5$ faster than QR decomposition-based SVD. We also tested the methods 50 times on $10^5, 10^6, 10^7$ and $10^8$ $2 \times 2$ matrices and have observed the speed of **Algorithm 1** is consistently $10^5$ faster than conventional SVD.

### A.3.2 PARAMETER SETTINGS OF RPSP

In this study, we set $T = 4$ and have validated this setting can accurately identify LLR submatrices in different scenarios. In this study, we set $N_1 =40$, $N_2 =200$, $N_3 =2000$, and $N_4 =6000$ that guarantee almost surely that the max cosine distance between any 2, 4, 8, and 16 dimension vector and the random unit vectors is larger than 0.98, 0.95, 0.92 and 0.8, respectively. We set the initial $L_1=10^7$, $L_t = \frac{L_{t-1}}{10}$, and we also accumulated the scoring matrix $S^t$ by adding $S^{t-1}$, and forgetting some scores with a random probability. This accumulation strategy can help our model learn from the previous iterations, and thus drastically reduce the running time and increase accuracy.

### A.3.3 PARAMETER SETTINGS OF BASELINE METHODS

RPSP is implemented by python 3.9.8 version and the python libraries numpy(1.19.0)Harris et al. (2020), pandas(1.4.1)pandas development team (2020), pytorch(1.6.0)Paszke et al. (2019),numba(0.50.1)Lam et al. (2015) etc.

The baseline methods includes SSVD Yang et al. (2014), SPCA Erichson et al. (2020), Plaid Lazzeroni & Owen (2002), CC Banerjee et al. (2007), LLORMA Lee et al. (2016). The first four methods were implemented in R environment. For SSVD, we used R package ssvd (version 1.0) and default parameters. For SPCA, we used R package sparsepca (version 0.1.2) and default parameters. For two bicluster methods Plaid and CC, we used R package biclust (version 2.0.1). Specifically for Plaid, we set 'background' parameter as True, 'fit.model' parameter as $y \sim m + a + b$ as tutorial. And for CC method, the parameter $\delta$ and $\alpha$ was set as 1.0 and 1.5 as instructed by the tutorial.

For LLORMA, we used the Global LLORMA from author's GitHub[1] with library Tensorflow-GPU 1.4.0. For synthetic data experiments, all experiments were conducted by setting PRE_RANK=1 along with other default parameters. In real-world data experiments, we set PRE_RANK = 10 along with other parameters in default. We used default learning rate parameter for all cases except for when testing the time consumption on input matrices of sizes $10000 \times 10000$ and $5000 \times 5000$, where the learning rate is set to be PRE_LARNING_RATE = 2e-5.

### A.3.4 EVALUATION METRIC OF SYNTHETIC DATA BASED EXPERIMENTS

For the simulated data. The Accuracy of detecting the true pattern and the back ground noise is used in our experiments to evaluate the performance of RPSP and the benchmarks. For one simulated input matrix $X^{M \times N}$, we labeled the true pattern as "1" and the back ground noise as "0". We keep tracking the index of the true pattern, thus we could generate a binary matrix as the ground-truth $GT^{M \times N}$. If we define the output from the methods above to be $X_{output}^{M \times N}$ and change the non-zero elements in $X_{output}^{M \times N}$ to "1". We could get the output binary matrix $GT_{output}^{M \times N}$. Then we can get the criterion of True Positive(TP), True Negative(TN), False Positive(FP), False Negative(FP) and the Accuracy by the following function:

$$TP, FP, FN, TN \leftarrow Confusion\_Matrix(GT_{output}^{M \times N}, GT^{M \times N}) \tag{1}$$

$$Accuracy \leftarrow \frac{TP + TN}{TP + TN + FP + FN} \tag{2}$$

Note that in some simulation settings, the method SPCA failed, so we gave its Accuracy of 0. In fact, all "0" Accuracy in the figures are because the methods could failed to give an output. For method CC, it sometimes detects the whole matrix as one pattern matrix, meaning that CC could not identify any of the true pattern. Under our evaluation metrics, CC will result in TP=1,TN=0,FP=0,FN=1, so we gave the Accuracy of 0.5.

### A.3.5 REAL-WORLD DATA PROCESSING

**MovieLens data:** We have gotten the permission from GroupLens to use the MovieLens datasetHarper & Konstan (2015) in our experiments and the datasets don't include sensitive information. MovieLens 25M data contains the ratings of 62,000 movies by 162,000 users. The MovieLens data is commonly used as a benchmark data for pattern detection. To ensure the rigor of evaluation, we selected the top 600 active users(rows) and the 600 most rated movies(columns) to build a testing dataset with a density rate of $20.0\%$. This is a relatively sparse dataset. Indeed, on high sparsity dataset, all baseline methods tend to identify LCV submatrices, but still RPSP is more favorable than others in terms of the low rankness and coverage rate of the detected patterns.

**Single Cell RNA-sequencing data:** Single cell RNA-sequencing (scRNA-seq) is a high throughput technique that measures the gene expression profile of individual cells Tirosh et al. (2016); Puram et al. (2017). The researchers are allowed to use the dataset in their study and the datasets don't

---

[1]https://github.com/JoonyoungYi/LLORMA-tensorflow

include sensitive information. The real application performed on two biomedical datasets, which are melanoma and head and neck cancer scRNA-seq data. We collected these two datasets from Gene Expression Omnibus (GEO) database, with accession ID GSE72056 and GSE103322. The cell type label and sample information provided in the original work were directly utilized. The GSE72056 data is collected on human melanoma tissues. The original paper provided cell classification and annotations including B cells, cancer-associated fibroblast (CAF) cells, endothelial cells, macrophage cells, malignant cells, NK cells, T cells, and unknown cells. The GSE103322 data is collected on head and neck cancer tissues. The original paper provided cell classification and annotations including B cells, dendritic cells, endothelial cells, fibroblast cells, macrophage cells, malignant cells, mast cells, myocyte cells, and T cells. We utilized a standard normalization protocol (FPKM) for both datasets, and selected the 4000 genes (rows) and 2000 cells (columns) with the highest expression values, bringing the density rates to $70.14\%$ (GSE72056) and $75.71\%$ (GSE103322). Notably, as indicated by the original work, malignant cells have high intertumoral heterogeneity. Theses two datasets provide us great opportunity to analysis the biological mechanism by identifying the local low rank pattern within data. And the total citation of two work is above 2000.

Both data sets have been utilized in more than 100 studies, in which GSE72056 contains 23684 genes and 4486 cell, and GSE103322 contains 22494 genes and 5902 cell. We utilize the standardized TPM measure of gene expression level as the input. Firstly, we first conducted a standardized normalization of the data by taking log+1: $X \leftarrow log_2(X + 1)$. We further selected the 4000 genes (rows) of top averaged expression level and the 2000 cells (columns) of the top total expression level to build our input testing data $X^{4000 \times 2000}$. The Low Rankness, Coverage Size(Size), Coverage Rate, Running Time were used as metrics in our experiments to evaluate the performance of RPSP and benchmark with other methods. For each submatrices calculated by the methods above, we compute its COR rate, Size(how many elements in it) and the coverage of rate of the submatrix to the input real data.

The rows represent the genes and the columns represent the cell. Each element in the matrix means the gene expression. The scRNA-seq data is sparse, the zero value in the matrix means the gene is not expressed in the cell. Thus, we just select 4000 rows and 2000 columns by their top mean value in our experiments. Specifically, the scRNA-seq data is the unstructured data, it isn't like the image data, the order of row or column doesn't have special mean. The experiments on scRNA-seq data, our goal is to get the local low rank submatrix from these data. The results (Fig S1) show that RPSP performances great to get the submatrices of LLR structure. The figures of Coverage Rate and Size show that RPSP can get the most obvious LLR structure submatrix and coverage enough submatrix size(product of number of row and column). The Running Time shows that RPSP has good time performance. RPSP consistently identified the local rank-1 pattern under this experimental setting.

**Spatial transcriptomics data:** 10x Genomics spatial transcriptomics (ST) is a recent commercialized technique to measure spatial coordinates associated gene expression signal from a biological tissue sample, and it has a huge utilization in biomedical studies. The researchers are allowed to use the dataset in their study and the datasets don't include personal information. We collected the spatial transcriptomics data human breast cancer tissue (v1.1 section 1) from https://www.10xgenomics.com/resources/datasets/, consisting 13161 genes and 3798 spatial spots. The data was processed and visualized by using Seurat 4.0 R packageHao et al. (2021). Counts data were directly utilized as the input of RPSP and other baseline methods. The row represents the genes and the column represents the cell. our goal is to get the local low rank submatrix form this data. The result(Fig S1e) shows one case of local low rank submatrix found by RPSP.

### A.3.6 EVALUATION METRICS OF REAL-WORLD DATA

Four evaluation metrics were utilized to evaluate the performance of each method, namely (1) the Low Rankness, evaluated by $\frac{\sigma_1}{||X_{I_k \times J_k}||_*}$ of an identified local low rank matrix $X_{I_k \times J_k}$: $\sum_{i \in I_k} PCC(X_{i,J_k}, V_{,1})$, where $\sigma_1$ is the largest singular value and $||X_{I_k \times J_k}||_*$ is the nuclear norm of $X_{I_k \times J_k}$, (2) the Size of a local low rank matrix, (3) the total Coverage Rate defined as the total number of entries in the top-$k$ identified patterns divided by the size of the input matrix, and (4) the Running Time. In addition, context specific meaning of the identified patterns were evaluated by prior knowledge on the row/column-wise features.

### A.3.7 COMPREHENSIVE SUMMARY OF REAL-WORLD DATA BASED EXPERIMENTS

Fig S1. Experiment on real-world data.

**Application on movieLens data:** MovieLens 25M data contains the ratings of 62,000 movies by 162,000 users provided by GroupLensHarper & Konstan (2015). The MovieLens data is commonly used as a benchmark data for pattern detection. To ensure the rigor of evaluation, we selected the top 600 active users(rows) and the 600 most rated movies(columns) to build a testing dataset with a density rate of 20.0%. This is a relatively sparse dataset. Indeed, on high sparsity dataset, all baseline methods tend to identify LCV submatrices, but still RPSP is more favorable than others in terms of the low rankness, coverage rate.

We evaluated the top significant local low rank matrices identified by each method. The low rankness of the top six significant local low rank matrices identified RPSP is consistently higher (∼0.8) than the ones detected by baseline methods (∼0.55) (**Fig S1a**), while the size of the patterns detected by RPSP is lower but at a similar level of the ones detected by LLORMA, SSVD and SPCA **Fig S1b**). Although the patterns detected by RPSP is slightly smaller, their coverage rate is higher than the results of LLORMA, SSVD and SPCA. This is because the ones detected by RPSP are distinctly non-overlapping while the top patterns identified by baseline methods are heavily overlapped (**Fig S1c**). On this data, Plaid did not detect any pattern while CC detected a large pattern formed by 309 users and 221 movies, whose low rankness is lower than the ones detected by RPSP. The RPSP has longer running time than LLORMA, SSVD and SPCA but is faster than CC. In sum, on the MovieLens data, FFLRM could detect distinct local low rank matrices of specifically high low rankness.

**Application on single cell RNA-seq data:** Single cell RNA-sequencing (scRNA-seq) is a high throughput technique commonly used in studying complex biological systems Tirosh et al. (2016); Puram et al. (2017). A typical scRNA-seq data is a matrix of ∼10,000 genes (rows) in ∼5,000 individual cells (columns) with a density rate in the range of 5% − 50%. The LCV and LLR submatrices in a scRNA-seq data directly correspond to sub-population of cells with distinct functions Wan et al. (2019a). We applied RPSP and SOTA methods on two real-world scRNA-seq data that are

most commonly utilized in testing pattern detection methods, namely GSE72056 (melanoma) and GSE103322 (head and neck cancer) Tirosh et al. (2016); Puram et al. (2017). Due to page limit, we only presented the results on GSE103322 in the main text. But here, we are presenting the results for both scRNA-Seq datasets. We utilized a standard normalization protocol (FPKM) and built the testing data by selecting the 4000 genes (rows) and 2000 cells (columns) with the highest expression values, bringing the density rates to 70.14% (GSE72056) and 75.71% (GSE103322).

On the two scRNA-seq data, both the low rankness and the size of the top patterns identified by RPSP are consistently higher than the ones detected by baseline methods (**Fig S1a-b**). The patterns detected by RPSP are much less overlapped than the ones detected by LLORMA and SSVD. RPSP also achieved the highest total coverage rate comparing to all baseline methods (**Fig S1c**). Plaid and CC only detected one local pattern, whose low rankness is much lower than the ones detected by RPSP, LLORMA and SSVD. The RPSP had longer running time than SSVD and SPCA but is faster than LLORMA and CC on the scRNA-seq data (**Fig S1c**). We also examined the biological meaning of the low rank patterns detected by RPSP, LLORMA and CC, by testing the enrichment of the gene features of each pattern against known biological pathways. Noted, only in RPSP, we found three local low rank matrices significantly enrich distinct biological functions including cell metabolism, cell proliferation, and antigen presentation. In summary, RPSP outperforms all baseline methods in detecting local low rank matrices on the scRNA-seq data sets, in terms of the low rankness, size, coverage rate and biological interpretability of detected patterns.

**Application on spatial transcriptomic data:** 10x Genomics spatial transcriptomics (ST) is a recent commercialized technique to measure spatial coordinates associated gene expression signal from a biological tissue sample, and it has been widely utilized in biomedical studies. A typical ST data is a matrix of ~15,000 genes (rows) in ~4,000 individual spatial spots (columns), and each spot has a 2D spatial coordinate (**Fig S1d**). A key challenge in ST data analysis is to infer the spatially dependent biological functional variations, which could be modeled as local low rank matrices formed by functionally associated genes over a certain spatial region, i.e. a MLLRR problem.

We applied RPSP and SOTA methods on the v1.1 ST data of breast cancer provided by 10xgenomics.com, consisting of 13161 genes and 3798 spatial spots with a density rate of 40.56%. Noted, as we have seen in the MovieLens and scRNA-seq data, RPSP is the only method that detected patterns of strong low-rankness. We showcased one of the LLR patterns specifically detected by RPSP (**Fig S1e**). The genes of this submatrix include two major groups, MHC class-I (immune signal given from cancers) and MHC class-II (immune signal received by immunes) antigen presenting genes. The spatial coordinates and signal level of this submatrix suggested the region of different levels of immune response in the cancer tissue (red regions in **Fig S1f** are of high immune response).

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
