# OpenReview forum: "GENERALIZED MATRIX LOCAL LOW RANK REPRESENTATION BY RANDOM PROJECTION AND SUBMATRIX PROPAGATION"
_ICLR.cc/2023/Conference — Submitted to ICLR 2023_

### Official Review · Reviewer_gxjs · 2022-10-28

**Confidence:** 3
**Correctness:** 2
**Technical Novelty And Significance:** 3
**Empirical Novelty And Significance:** 3
**Recommendation:** 3

**Clarity, Quality, Novelty And Reproducibility:**

- The paper is not well written. It is difficult to understand in many places because of the writing style, grammatical errors, and confusing language.
- While the algorithm presented in the paper appears to be somewhat novel, many individual aspects of the algorithm are well known in the literature.
- The paper does not point to any publicly available repo, so the reproducibility of the paper cannot be evaluated.

**Strength And Weaknesses:**

**Strengths**

- The algorithm, while mostly heuristic in nature, seems to be effective in finding the low-rank submatrices, at least on the matrices reported in the paper.
- There are numerical experiments that showcase the workings of the algorithm in comparison to other approaches in the literature.

**Weaknesses**

- The paper uses confusing and imprecise terminology, which makes it hard to pinpoint the main problem being studied in the paper. For example, the terminology of *low constant variation (LCV)* and *local linear low rank (LLR)* is never defined in a mathematically precise manner. The mathematical formulation in the paper never clarifies whether the indices $I_k \times J_k$ are meant to be disjoint or not. The concepts of *background noise*, *background error*, etc., are never defined concretely and it is also not clear if this is a requirement or not. In the absence of this mathematical rigor, one wonders if the problem is even a well posed one, and this is the biggest weakness of this paper.
- Lemma 1 seems to be gratuitous in nature. It is given under the assumption that $N^R \rightarrow \infty$, which makes it trivial, and it does not add anything to our understanding of the algorithm.
- Many of the ideas being presented in the paper, such as the use of random projections to approximate the minimum and maximum singular values of a matrix, are well known in the literature, but the paper does not make this clear.

**Summary Of The Paper:**

The focus of this paper is on finding smaller-sized low-rank submatrices within a larger matrix. The main contribution of the paper is an algorithm that uses random projections along with a few other steps to find the low-rank submatrices. There is also a minor theoretical result in the paper that is asymptotic in nature.

**Summary Of The Review:**

The results presented in the paper seem to be effective, but the paper needs a lot of work to make it publishable.

---

> ### Author Response · Authors · 2022-11-05
> **Response from authors**
>
> We appreciate the reviewer's time and effort in reviewing our study. The reviewer expressed concern about the mathematical correctness of our method. We want to note these issues raised by the reviewer can be easily addressed and the noise-related concern is unnecessary for the proposed study. We wish the reviewer could reconsider the mathematical aspect of our study within the scheme of matrix low-rank representation. And we wish the reviewer can recognize the originality of this study. We believe the mathematical consideration, formulation of the solution, and performance of our proposed method show substantial novelty and significance.
>
> Point-to-point responses:
>
> Concern: The paper uses confusing and imprecise terminology, which makes it hard to pinpoint the main problem being studied in the paper. For example, the terminology of low constant variation (LCV) and local linear low rank (LLR) is never defined in a mathematically precise manner.
>
> Response: As discussed in our manuscript, matrix local low-rank representation (MLLRR) is not a well-solved problem. As the past methods can only solve the LCV problem, the LLR problem was very little studied. Here we first provide a mathematical formulation of these two MLLRR sub-problems.
>
>
> Concern: The mathematical formulation in the paper never clarifies whether the indices I_k*J_k are meant to be disjoint or not.
>
> Response: Its definition was provided in section 2.1. The indices I_k*J_k can be any subset of indices, which are meant to be disjoint.
>
>
> Concern: The concepts of background noise, background error, etc., are never defined concretely and it is also not clear if this is a requirement or not.
>
> Response: We appreciate the reviewer raised this issue. However, SVD, nuclear norm, PCA, and other low-rank representation frameworks of a matrix are not based on any pre-assumed background noise or error, as these frameworks reflect the underlying low-rankness property of a matrix, which is independent of noise types. In addition, the random projection-based singular value computation and even statistical inference of the rankness of a matrix do not rely on pre-assumed background noise type. We have discussed background errors in section 2.1-2.2, and mentioned "Of note, in this study, we do not restrict the form of the background noise distribution.".
>
> Ref: [1] Owen, A. B., & Perry, P. O. (2009). Bi-cross-validation of the SVD and the nonnegative matrix factorization. The annals of applied statistics, 3(2), 564-594. Liu, [2] Guangcan, et al. "Robust recovery of subspace structures by low-rank representation." IEEE transactions on pattern analysis and machine intelligence 35.1 (2012): 171-184.
>
>
> Concern: In the absence of this mathematical rigor, one wonders if the problem is even a well-posed one, and this is the biggest weakness of this paper.
>
> Response: We respectfully disagree with the reviewer. We are confident of the mathematical rigorousness of this study, as the key computational steps are supported by random projection and sigma-packing theories.
>
>
> Concern: Lemma 1 seems to be gratuitous in nature. It is given under the assumption that N^R -> infinity, which makes it trivial, and it does not add anything to our understanding of the algorithm.
>
> Response: The purpose of citing Lemma 1 is just to provide theoretical support for the correctness and rationale of our method. It suggests that random projection can effectively compute the singular value. This lemma is not considered a key contribution of this work.
>
> Concern: Many of the ideas being presented in the paper, such as the use of random projections to approximate the minimum and maximum singular values of a matrix, are well-known in the literature, but the paper does not make this clear.
>
> Response: Our method is based on the well-studied random projection. We cited related works. We will thoroughly discuss existing studies of random projection in our revision. However, the goal of this study is to solve the unaddressed MLLRR problem. To the best of our knowledge, RPSP is the first time utilization of random projection to solve the MLLRR problem. We hope the reviewer can recognize the true novelty and contribution of this work on the MLLRR problem.
>
>
> Concern: The paper is not well written. It is difficult to understand in many places because of the writing style, grammatical errors, and confusing language.
>
> Response: Considering the reviewers can be of different backgrounds, we presented the main algorithm and consideration in a more descriptive way and put detailed explanations in the APPENDIX. We will substantially improve our writing in the revised version. We want to re-emphasize the importance and challenges of the MLLRR problem and the performance of our method.
>
>
> Concern: The paper does not point to any publicly available repo, so the reproducibility of the paper cannot be evaluated.
>
> Response: We want to note that the ICLR guidelines prohibit the deposition to a publicly available repo.

---

> > ### Comment · Reviewer_gxjs · 2022-12-11
> > **Reviewer's response to authors' response**
> >
> > I would like to thank the authors for their response and I regret the fact I could not engage with them earlier during the discussion period because of personal reasons. I have checked the system and it does not appear the authors have uploaded a revised version of the paper. I have also gone through the authors' response below and, in the absence of a revised version, I am unable to change my score for the paper. I encourage the authors to take into account the provided feedback and improve their presentation / strength of results for future submissions.
> >
> > In response to the authors' response, my brief comments are provided below.
> >
> > > The reviewer expressed concern about the mathematical correctness of our method ... And we wish the reviewer can recognize the originality of this study. We believe the mathematical consideration, formulation of the solution, and performance of our proposed method show substantial novelty and significance.
> >
> > The method appears to be mathematically correct. My concern is the mathematical formulation of the problem in a non-rigorous manner, as well as the presentation that is imprecise in many places. While the authors state that this "can be easily addressed", no revision is provided that can convince me of this.
> >
> > > Response: As discussed in our manuscript, matrix local low-rank representation (MLLRR) is not a well-solved problem. As the past methods can only solve the LCV problem, the LLR problem was very little studied. Here we first provide a mathematical formulation of these two MLLRR sub-problems.
> >
> > The authors are repeating the phrasing from the paper. My concern is that many concepts are not *precisely* defined in the paper in a mathematically rigorous manner.
> >
> > > Response: Its definition was provided in section 2.1. The indices I_k*J_k can be any subset of indices, which are meant to be disjoint.
> >
> > I searched the paper again and could not find the word "disjoint" in there. I humbly request the authors to focus on the feedback in the reviews. The authors say "which are meant to be disjoint", but that's my concern. There are many things in the paper that are perhaps known to the authors, but which are not clearly stated in the paper.
> >
> > > Response: We appreciate the reviewer raised this issue ... We have discussed background errors in section 2.1-2.2, and mentioned "Of note, in this study, we do not restrict the form of the background noise distribution.".
> >
> > The issue is not on the "assumption" of noise in prior works. The concern  / issue is that things are not properly discussed / stated in the paper.
> >
> > > Response: We respectfully disagree with the reviewer. We are confident of the mathematical rigorousness of this study, as the key computational steps are supported by random projection and sigma-packing theories.
> >
> > In the absence of a revision, I unfortunately do not agree with the authors' view that the paper is well written.
> >
> > > Response: The purpose of citing Lemma 1 is just to provide theoretical support for the correctness and rationale of our method. It suggests that random projection can effectively compute the singular value. This lemma is not considered a key contribution of this work.
> >
> > As stated in the review, the lemma in fact does not provide the theoretical support, as it is asymptotic in nature and the result is well known in the literature.
> >
> > > Response: Our method is based on the well-studied random projection. We cited related works. We will thoroughly discuss existing studies of random projection in our revision ... We hope the reviewer can recognize the true novelty and contribution of this work on the MLLRR problem.
> >
> > The authors have indeed cited related works, but the contribution of this work is limited in light of those related works.
> >
> > > Response: Considering the reviewers can be of different backgrounds, we presented the main algorithm and consideration in a more descriptive way and put detailed explanations in the APPENDIX. We will substantially improve our writing in the revised version. We want to re-emphasize the importance and challenges of the MLLRR problem and the performance of our method.
> >
> > Since I do not see a revised version, I cannot change my score.
> >
> > > Response: We want to note that the ICLR guidelines prohibit the deposition to a publicly available repo.
> >
> > Publicly available repo versus publicly identifiable repo are two different things. The authors can check that many ICLR submissions have anonymized publicly available repos for their code. This however is a minor comment and was put there since it is part of ICLR review. It is not the reason for my score.

---

> ### Author Response · Authors · 2022-11-18
> **We are looking forward to your further comments**
>
> Dear reviewer gxJs,
>
> We must admit that studying the np-hard problem and the associated estimation techniques is clearly much more difficult than improving already proven techniques. However, we believe that our approach does provide a workable direction in this challenging set of problems. In order to contribute to the study of such difficult fundamental problems, we will also continue to improve our approach in our future work.
> I want to thank you again for reading our work carefully and leaving thoughtful comments. They are really important to enhance our work. The deadline for discussing Phase I is fast approaching and we welcome your comments and/or inquiries. We truly believe that our rebuttals have sufficiently allayed your concerns. If so, we sincerely hope that you will improve your score. If not, please let us know of your further concerns and we will continue to actively address your questions and improve our contribution.
>
> Best,
> Authors

---

### Official Review · Reviewer_8ATp · 2022-10-30

**Confidence:** 4
**Correctness:** 3
**Technical Novelty And Significance:** 2
**Empirical Novelty And Significance:** 2
**Recommendation:** 3

**Clarity, Quality, Novelty And Reproducibility:**

Parts of the paper are poorly written and it is hard to follow some of the ideas introduced. Namely, Section 3.2, which introduces the main methodology of the paper should be presented in more detail. Both the submatrix Propagation step and the low-rank prediction step  omit significant information that would allow the reader to better understand the proposed methodology.

**Strength And Weaknesses:**

Strengths:

- The paper deals with a challenging and interesting problem i.e., matrix local low-rank representation.
- The authors provide a method, which is based on random projections and that helps towards deriving a more efficient method relative to other approaches.
- The authors provide numerous simulated and real-data experiments for evaluating their approach.

Weaknesses:
 - The main idea of the paper is the use of the random projection-based method for estimating the singular values of the submatrices in an efficient way. The authors also present Lemma 1 to support that singular value approximation method. However, they seem to ignore the literature and the methods already developed for randomized SVD [1,2]. The random projection-based method presented in Section 3.1, claimed by the authors as a novel contribution of the current work, relies on already existed work ([1,2]) that is not cited in the paper.
- Lemma 2 is poorly explained in the manuscript and hence is hard to grasp its contribution and impact of it in the paper. Can the authors use it in practice the result to get an upper bound on the number of submatrices they will need to consider?
- Section 3.2 is also poorly presented, and several steps of the methodology followed and given in Figure 3 are omitted or lack detailed explanation. For instance, it is not clear how the weighted sampling is done and how the local low-rank submatrices are determined using the scoring matrix.  Since these steps are critical to the method, further details should be provided in the main paper. Moreover, what is the computational complexity of the co-clustering algorithm that is used, and how that affects the efficiency of the whole approach?


Minor Comments

- Truncated SVD shouldn't be given as Definition 1.
- In the random projection-based algorithm, what is the sensitivity in the selection of the cut-off angle \theta?

[1] Halko, Nathan, Per-Gunnar Martinsson, and Joel A. Tropp. "Finding structure with randomness: Probabilistic algorithms for constructing approximate matrix decompositions." SIAM review 53.2 (2011): 217-288.
[2] Martinsson, Per-Gunnar, and Joel Tropp. "Randomized numerical linear algebra: Foundations & algorithms." (2021).

**Summary Of The Paper:**

The authors propose a method for local low-rank representation of matrices. The main idea is initially sample small size submatrices that progressively grow. At each step, a random projection-based method is used to evaluate the low-rankness of each matrix and assigning a score, which reflects the probability that a small-size matrix is contained in a larger one. These scores form a scoring matrix which determines the final local low-rank matrices that represent the input data matrix. The authors evaluate their method on simulated and real datasets such as Movielens and RNA sequencing data showing the promising performance of their method in terms of accuracy of the approximations and time efficiency of their method.

**Summary Of The Review:**

The authors present a methodology for matrix local low-rank representation. The main idea is to use a random projection-based method which allows them to efficiently derive low-rankness scores for progressively growing submatrices. These are used to derive the local low-rank representation. The author's main contribution relies on ideas of randomized linear algebra and prior work, which is not cited in the paper by the authors. Moreover, the main idea are poorly explained (see above) it is not clear to assess the significance of the results and that motivate the proposed methodology.

-----------------------------------------
Post-Rebuttal Update:
I appreciate the authors' effort to respond to reviewers' comments and concerns, and I would like to thank them for that. However, this time I will not change my score. I think the paper's current version needs significant changes to meet the standards of ICLR. I encourage the authors to improve their manuscript based on the reviewers' suggestions and resubmit it to another conference.

---

> ### Author Response · Authors · 2022-11-05
> **Response from authors: part 1**
>
> We appreciate the Reviewer's time and effort in reviewing our manuscript. We appreciate that the reviewer identified that we are dealing with a challenging and interesting problem. After carefully reading your review comments, we consider the weaknesses are highly addressable. First, we want to clarify that the focus of this work is to develop a feasible approach to identify local low-rank sub-matrices from a big matrix, especially for the local matrices having a similar mean to the background or having non-Gaussian errors. We never wished to claim that the novelty and significance of this work is the random projection-based Singular Value computation. Instead, our method was based on this idea, and to the best of our knowledge, this is the first approach that implements this idea for local low-rank sub-matrices identification. We fully respect the previous studies of random projection-based singular value computations.
>
> Below are our point-to-point responses to the reviewer:
>
> Concern 1: The main idea of the paper is the use of the random projection-based method for estimating the singular values of the submatrices in an efficient way. The authors also present Lemma 1 to support that singular value approximation method. However, they seem to ignore the literature and the methods already developed for randomized SVD [1,2]. The random projection-based method presented in Section 3.1, claimed by the authors as a novel contribution of the current work, relies on already existed work ([1,2]) that is not cited in the paper.
>
> Response: We want to note that the purpose of citing Lemma 1 and Lemma 2 is simply to provide theoretical bases for our proposed method. We fully respect the previous studies of random projection in computing matrix singular values by using random projection. We hope to convey to the reviewer our key contribution as follows: (1) we developed a method based on random projection-based singular value computation, which has been well studied, to address the local low-rank matrix identification problem, which was not well solved by the existing methods; (2) although our RPSP adopts the idea of random projection, to the best of our knowledge, it is still the first method that can solve the general matrix low local rank representation problem. RPSP outperforms the existing method. It can identify a local low-rank matrix when the mean of the submatrix is similar to the background's and it does not rely on any assumption of background noise. Our synthetic data revealed that the baseline methods cannot identify any pattern under multiple experimental conditions while RPSP is robust to all experimental settings. (3) Our RPSP screens each potential local low-rank matrix via visiting randomly sampled submatrices providing a new perspective for solving the local low-rank matrix detection problem. All existing methods utilize a global screening (with sparse constraints) or anchor-based screening of local patterns.  In our mathematical analysis A.2.1 in the Appendix, we discussed that if the size local low-rank matrix is small and the mean of its values is not distinct compared to the background, no global screening method can identify the submatrix.
>
> We want to note that the two references provided by the reviewer are for random projection-based singular value estimation of a global matrix rather than local matrices. We have cited a similar study "N Benjamin Erichson, Peng Zheng, Krithika Manohar, Steven L Brunton, J Nathan Kutz, and Aleksandr Y Aravkin. Sparse principal component analysis via variable projection. SIAM Journal on Applied Mathematics, 80(2):977–1002, 2020.". We will revise our manuscript to provide a more comprehensive review of the random projection-based method for estimating the singular values. We will remove the key contribution (4) to avoid further misunderstanding.
>
> We hope the above discussion can clarify that we never wanted to exaggerate our contributions and we are confident about the true contribution of this work.
>
>
> Concern 2: Lemma 2 is poorly explained in the manuscript and hence is hard to grasp its contribution and impact of it in the paper. Can the authors use it in practice the result to get an upper bound on the number of submatrices they will need to consider?
>
> Response:
> We apologize for the less discussion of Lemma 2, as it was adopted from a previous study. Lemma 2 helps to determine the minimum number of unit vectors that are needed to ensure a compact coverage to an R-dimensional space. In our RPSP method, we determined the number of randomly generated unit vectors for the dimensions 2, 4, 8, and 16 based on Lemma 2. We have mentioned that "Algorithm 1: Singular Value Approximation. Its input includes a matrix X (Fig 2a) and a set of randomly generated unit vectors P (Fig 2b), whose cardinality is bounded by Lemma 2." We will provide more discussion of Lemma 2 in our revised version.

---

> ### Author Response · Authors · 2022-11-05
> **Response from authors: part 2**
>
> Concern 3: Section 3.2 is also poorly presented, and several steps of the methodology followed and given in Figure 3 are omitted or lack detailed explanation. For instance, it is not clear how the weighted sampling is done and how the local low-rank submatrices are determined using the scoring matrix. Since these steps are critical to the method, further details should be provided in the main paper.
>
> Response:
> We apologize for the unclear presentation of Section 3.2. Actually, detailed algorithms and descriptions of weighted sampling and local low-rank submatrices determination were given in Appendix. We focused on presenting the key steps and considerations of our RPSP algorithm in Section 3.2 and provided more explanation of sub-algorithms in the Appendix. We will substantially revise the description of Section 3.2 to clarify each sub-algorithms.
>
> Concern 4: Moreover, what is the computational complexity of the co-clustering algorithm that is used, and how that affects the efficiency of the whole approach?
>
> Response:
> As mentioned in the Appendix, we utilized a Spectral Co-Clustering approach, which has a relatively low computational cost. Because the computational cost of our RPSP method relies on the parameter setting and the GPU machine, hence instead of providing a theoretical analysis of the scalability, we provide the running time of the singular value estimation step under different parameter settings (Table 3 in the Appendix). Considering a large number of screened random matrices, the singular value estimation step is the most time-consuming step compared to all other steps. We also examined the running time of RPSP vs baseline methods in all experiments. We will provide a more detailed discussion and analysis of the scalability of our method in the revised version.
>
>
> At last, we want to thank the reviewer again. And we want to re-emphasize that our RPSP algorithm effectively solves the un-fully addressed local low-rank matrix identification problem. Although PRSP is a heuristic algorithm, its mathematical consideration and key computational steps are supported by mathematical theories of random projection and random packings and coverings of the unit n-sphere.

---

> ### Author Response · Authors · 2022-11-18
> **We are looking forward to your further comments.**
>
> Dear reviewer 8ATp,
>
> We must admit that studying the np-hard problem and the associated estimation techniques is clearly much more difficult than improving already proven techniques. However, we believe that our approach does provide a workable direction in this challenging set of problems. In order to contribute to the study of such difficult fundamental problems, we will also continue to improve our approach in our future work.
> I want to thank you again for reading our work carefully and leaving thoughtful comments. They are really important to enhance our work. The deadline for discussing Phase I is fast approaching and we welcome your comments and/or inquiries. We truly believe that our rebuttals have sufficiently allayed your concerns. If so, we sincerely hope that you will improve your score. If not, please let us know of your further concerns and we will continue to actively address your questions and improve our contribution.
>
> Best,
> Authors

---

### Official Review · Reviewer_8ysf · 2022-11-04

**Confidence:** 4
**Clarity, Quality, Novelty And Reproducibility:** 1. This paper is well organized. Expe…
**Correctness:** 3
**Technical Novelty And Significance:** 3
**Empirical Novelty And Significance:** 3
**Recommendation:** 5

**Strength And Weaknesses:**

Strength

1. RPSP is supported by rigorously derived mathematical theories, e.g. Lemma 1 and 2.

2. RPSP is the first method capable of handling the general MLLRM problem.

3. In the experiments, RPSP outperforms all baseline methods.

4. Experimental settings and parameter settings are clear.

Weaknesses

1. Some mathematical expressions are difficult to understand. For examples, a few commas are missing in “k = 1...K” and other places. Corner signs almost always use superscripts, and lack the necessary explanation, e.g., does “N^R” denote N to the power of R? Does “10^-5 - 10^-2” means subtracting two numbers? And I cannot understand “for L^1 = 10^{1}12×2 matrices”.

2. P is randomly generated, which is inefficient.

3. As shown in Figure 4 and Table 1, even computing parallelly, RPSP has little advantage in running time.

4. Lack of parameter sensitivity analysis, such as the experiments on the sensitivity of $\theta$.



**Summary Of The Paper:**

This paper provides a new computational framework of the matrix local low rank representation (MLLRR), namely Random Probing based submatrix Propagation (RPSP). RPSP is the first method capable of handling the general MLLRM problem. Specifically, RPSP utilizes a random projection and GPU-based computation of singular values for a large set of small matrices. It propagates the low rankness identified from small matrices to larger ones to identify local low rank submatrices of coherent patterns. On both synthetic and real-world experiments, RPSP outperforms all baseline methods on the LLR problems for data of different sparsity level and error distributions.

**Summary Of The Review:**

The authors develops  a sub-matrix propagation based approach to solve the fundamental mathematical problem of matrix local low rank representation.  This paper is well organized. Experimental settings and parameter settings are clear.  However, some details in the experiments need further clarification.

---

> ### Author Response · Authors · 2022-11-18
> **We are looking forward to your further comments.**
>
> Dear reviewer 8ysf,
>
> We must admit that studying the np-hard problem and the associated estimation techniques is clearly much more difficult than improving already proven techniques. However, we believe that our approach does provide a workable direction in this challenging set of problems. In order to contribute to the study of such difficult fundamental problems, we will also continue to improve our approach in our future work.
> I want to thank you again for reading our work carefully and leaving thoughtful comments. They are really important to enhance our work. The deadline for discussing Phase I is fast approaching and we welcome your comments and/or inquiries. We truly believe that our rebuttals have sufficiently allayed your concerns. If so, we sincerely hope that you will improve your score. If not, please let us know of your further concerns and we will continue to actively address your questions and improve our contribution.
>
> Best,
> Authors

---

> ### Author Response · Authors · 2022-11-19
> **Response from authors**
>
> We appreciate the reviewers' time and effort in handling our manuscript. We appreciate the reviewer identified that the problem is hard and important. We consider the reviewer's negative comments are minor and can be easily solved
>
> Below please see our point-to-point response to reviewer's concern.
>
> Concern 1: Some mathematical expressions are difficult to understand. For examples, a few commas are missing in “k = 1...K” and other places. Corner signs almost always use superscripts, and lack the necessary explanation, e.g., does “N^R” denote N to the power of R? Does “10^-5 - 10^-2” means subtracting two numbers? And I cannot understand “for L^1 = 10^{1}12×2 matrices”.
>
> Response: k=1...K means k takes integer values from 1 to K. The definition of superscripts is given in 2.1. Without a specific note, all representations and symbols follow general mathematical representations. N^R denote N to the power of R and 10^-5 - 10^-2 means in the range from 10^-5  to 10^-2, and L_T 2^T ×2^T matrices mean in total the number of 2^T ×2^T matrices is L_T. We consider these to be all general mathematical language and do not need to be specified in a scientific article.
>
> Concern 2: P is randomly generated, which is inefficient.
>
> Response: Randomly generated P is an unbiased inquiry of potential linear bases. And the procedure can be supported by the theory of random projection. Prefixed P may cause a certain level of bias.
>
> Concern 3: As shown in Figure 4 and Table 1, even computing parallelly, RPSP has little advantage in running time.
>
> Response: We do not consider the running time is an advantage of RPSP. We want to note that RPSP is the first method that can well solve the MLLRR problem when the mean of the low-rank pattern is similar to the background. Also the method is robust to different types of background errors.
>
> Concern 4: Lack of parameter sensitivity analysis, such as the experiments on the sensitivity of theta
>
> Response: We appreciate the reviewer's consideration of the sensitivity of theta. We have conducted an extensive evaluation of the robustness of RPSP and its sub-algorithms with respect to each hyperparameter. Theta in algorithm 1 is pre-given by users, which could be determined based on the number of samples on random bases. We have validated that the theta ensures cos(theta) > 0.1 can consistently achieve good accuracy in predicting singular values. However, smaller theta demands more randomly sampled bases. Our experiment suggests that the current setting of RPSP is not sensitive to theta. We will include this data in our revised version.

---

### Decision · Program_Chairs · 2023-01-20

**Decision:**

Reject

**Justification For Why Not Higher Score:**

The reviewers all recommended rejection.

**Justification For Why Not Lower Score:**

N/A

**Metareview: Summary, Strengths And Weaknesses:**

The authors propose a method for progressively growing local low-rank representation of matrices using random projections. The evaluation shows that the method has promising accuracy and time efficiency. Reviewers agreed that although the technical content seems plausible, the paper in its current form is difficult to understand, the experimental methodology is unclear, and clear connections and attributions to relevant prior results in the literature are missing.